# Adapting a Generative Pretrained Transformer Achieves SOTA Performance in Assessing Diverse Physiological Functions Using Only Photoplethysmography Signals: A GPT-PPG Approach

**Zhaoliang Chen[1], Cheng Ding[2], Nirbhay Modhe[3], Jiaying Lu[1], Carl Yang[1], Xiao Hu[3]**

[1]Department of Computer Science, Emory University
[2]Department of Biomedical Engineering, Georgia Institute of Technology
[3]Nell Hodgson Woodruff School of Nursing, Emory University

## Abstract

This study introduces a novel application of a Generative Pretrained Transformer (GPT) model tailored for photoplethysmography (PPG) signals, serving as a foundation model for various downstream tasks. Adapting the standard GPT architecture to suit the continuous characteristics of PPG signals, our approach demonstrates promising results. After pre-training on our extensive dataset that contains more than 200 million 30s PPG samples, the model shows performance comparable to or surpassing current state-of-the-art (SOTA) methods in tasks like heart rate estimation. A standout feature of our GPT model is its inherent capability to perform generative tasks such as signal denoising effectively, without the need for further finetuning. This success is attributed to the generative nature of the GPT framework. Looking ahead, we aim to further explore its generative abilities and investigate its implication on its other downstream tasks.

## Introduction

The emergence and success of large language models (LLMs) like BERT (Delvin et al. 2019) and GPT (Radford et al. 2018) have revolutionized our understanding of foundation models in artificial intelligence. These models, characterized by their extensive pretraining on large datasets without explicit supervision, demonstrate remarkable versatility across a range of downstream tasks via finetuning. This concept of foundation models has particularly significant implications in the realm of clinical data analysis. In clinical settings, the challenge often lies in the limited availability of labeled training samples. Foundation models, with their inherent flexibility and generalizability, offer a promising solution to this bottleneck.

The architecture and training methodologies of foundational models in natural language processing provide a template that can be adapted to other domains. BERT's approach, centered on masked word reconstruction and next sentence prediction, contrasts with GPT's focus on predicting the next token in a sequence. This sequential processing capability opens up intriguing possibilities for the analysis of time-series data in healthcare, such as Photoplethysmography (PPG) signals. PPG signals, akin to sequences of continuous tokens, are ripe for exploration through the lens of NLP-inspired models.

In this work, we introduce a novel PPG foundation model, utilizing the GPT architecture as our base. Our model is pretrained on a dataset comprising more than 200 million 30-second PPG signals. While adhering to many of the design choices common to widely recognized open-source LLMs, such as Llama from Meta, we have innovatively adapted the embedding and output layers to cater specifically to the characteristics of PPG signals.

The contributions of this study are manifold. We demonstrate the robust capabilities of our model across various downstream tasks, including classification, regression, and generative applications. We also provide insights into how we may adapt techniques from NLP to the domain of continuous time series data.

## Related Work

Inspired by the transformative success of the transformer architecture in capturing long-range dependencies, several works have adapted this framework for time series analysis. PatchTST, introduced by Nie et al. (2023), innovatively treats each time series patch as a discrete token, training a feature extractor through masked reconstructions. Zerveas et al. (2020) diverged from this approach by considering each timestamp as the fundamental unit of information, employing a transformer encoder as the backbone. While these methods leverage the encoder architecture, there's a growing interest in utilizing the transformer decoder, aligning more closely with the GPT model's predictive capabilities. Notably, EarthPT (Smith, Fleming, Geach, 2024) adopts a GPT-like structure for Earth Observation data, treating each observation as a token. Our model distinguishes itself by treating patches as tokens, similar to PatchTST, which offers computational efficiency and potentially richer local feature

representations due to the reduced sequence length inherent in our approach.

The exploration of PPG signals through deep learning methodologies, such as the convolutional recurrent regressor by Ismail et al. (2022) for heart rate estimation and the novel deep recurrent network by Chowdhury et al. (2020) for blood pressure estimation, underscores the potential of advanced models in extracting meaningful features from PPG data. However, the application of transformer-based models, particularly with adaptations from the GPT architecture, presents a novel frontier in enhancing the accuracy and efficiency of PPG signal analysis.

# Method: GPT Architecture and Pretraining

In leveraging the Generative Pre-trained Transformer (GPT) framework for analyzing continuous time-series PPG signals, we preserved the core components of GPT while making key adjustments to suit the unique characteristics of physiological signals. The decision to employ GPT was driven by its proven capability in capturing complex dependencies within sequential data, making it a promising candidate for time-series analysis.

Our model was pretrained on a dataset containing 30s PPG signals, all resampled to 40Hz. Prior to feeding PPG signals into the model, we first normalize each sample $X \in \mathbb{R}^{1200}$ into $[0,1]^{1200}$ by min-max normalization. That is:

$$X = \frac{X - \min(X)}{\max(X) - \min(X)}$$

Then we reshape the normalized signal into 30 consecutive, non-overlapping patches, $X = \{x_1, x_2, \ldots, x_{30}\}$, with each patch $x_i \in \mathbb{R}^{40}$ encapsulating 1 second of the signal. The code, model weights and training hyperparameters will later be made publicly available.

We trained two models, GPT-19M and GPT-85M, where the former is only trained on 5% of the entire dataset due to the cost. GPT-85M was trained for 5 epochs, each taking approximately 45 hours on 4 A6000 GPUs.

## Embedding

The embedding layer serves a pivotal role in our model, transforming each patch into a high-dimensional vector representation. Our model uses a simple yet effective linear layer to map each patch to a vector of dimension $d_m$.

In the standard GPT implementation, we typically shift the entire sequence to the right by injecting a special token such as [SOS] in front to denote start of sentence and dropping the last token. We employ a similar technique. However, instead of injecting special numbers in front of the PPG sequence, a learnable vector $h_s$ of dimension $d_m$ is registered to the model directly and prepended to the embedded PPG sequence. Therefore, the whole embedded PPG sequence is $\{h_s, h_1, h_2, \ldots, h_{29}\}$. This approach not only

facilitates the model's training by providing a consistent starting point for each sequence but also helps mitigate the issue of out-of-distribution patches that could otherwise impair the linear embedding layer's effectiveness. Since the signal is always 30s long in the pretraining phase, the traditional [EOS] (end of sequence) token is rendered unnecessary.

## Stacked Transformer Decoder

Our stacked transformer decoder layer follows the common design choices. Prior to attention module and feed-forward network (FFN), we employ root mean square normalization (RMSNorm), introduced by Zhang and Sennrich (2019), instead of layer normalization (LN). This choice is motivated by RMSNorm's potential to enhance model training dynamics. In addition, we use rotary positional embedding (RoPE), introduced by Su et al. (2023), to capture relative positional information of the sequence.

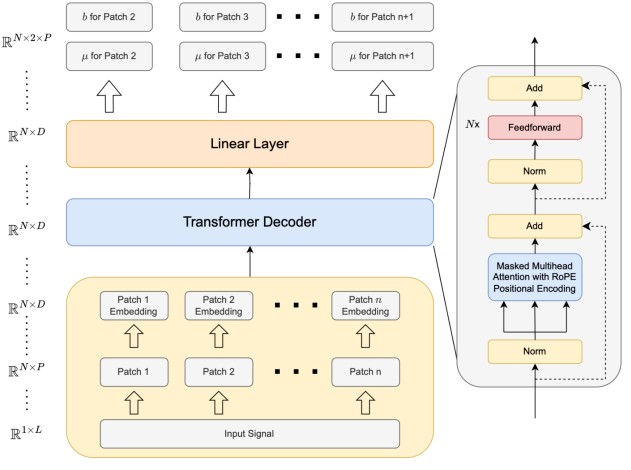

Figure 1: Diagram Illustrating GPT Model Design

## Prediction Head and Loss Function at Pretraining

In adapting the GPT architecture for continuous PPG signal prediction, our model's prediction head diverges from the typical categorical output seen in NLP tasks. This necessitates a tailored approach to both the prediction mechanism and the choice of loss function.

A natural choice is to directly map hidden vectors back to the input space and use a distance loss function like MSE. However, our initial experiments with MSE led to model collapse, where the model predominantly predicts the mean value of the normalized signals, converging to a non-informative constant around 0.5. We conjecture that the cause of this issue is the fact that MSE has support over the entire real line while the signal is normalized to $[0,1]$ interval, which poses a significant distribution mismatch.

To address this issue, we use a distribution-based loss function that has support of (0, 1), called logit-Laplace distribution loss (Ramesh et al. 2021). Instead of predicting the input signal, the model predicts vectors $\mu, b$, each having the same shape as input, representing the location and scale parameters of the distribution.

The logit-Laplace distribution is given by applying a sigmoid transformation over a Laplace distributed random variable. Its probability distribution function (PDF) is given by:

$$f(x|\mu, b) = \frac{1}{2bx(1-x)} \exp\left(-\frac{|\text{logit}(x) - \mu|}{b}\right)$$

where $x$ is ground truth, $\mu$ is the predicted location parameter and $b$ is the predicted scale parameter. It can be shown that minimizing the negative log likelihood of a Laplace distribution is equivalent to minimizing L1 distance loss, and applying the sigmoid transformation restricts the support to (0, 1) open interval, making the negative log of the above PDF a suitable loss function.

However, the normalization of signals to the [0, 1] interval poses a challenge at the boundaries. Due to the nature of the min-max normalization, we are guaranteed that 0 and 1 will occur in each 30s sample, and either will cause the denominator $2bx(1 - x)$ go to 0. To avoid this numerical issue, we perform an invertible transformation on $x$ to ensure that it is in [0.1, 0.9] interval. This approach is favored over adding a small constant $\epsilon$ to the denominator, which will likely cause instability in training because then the term $1/\epsilon$ will occur in the loss function for every training sample.

## Method: Supervised Finetuning

A key advantage of the GPT architecture is its adaptability to varying sequence lengths. This attribute is particularly beneficial in finetuning, where the sequence length may be different from the 30-second segments used in pretraining, allowing for seamless adjustment to the specific requirements of each task.

We explored 2 types of downstream heads, one linear and the other attention based. In the finetuning process, we defined our loss function as a linear combination of the objective loss $L_o$, corresponding to the finetuning objective, and signal modeling loss $L_m$, which is logit-Laplace distribution loss obtained on the downstream dataset, similar to the combined loss function proposed by Radford et al. (2018).

### Linear Prediction Head

In the finetuning phase for downstream tasks, our model employs a linear prediction head to transform the output of the last transformer decoder block, which has dimension (sequence length, $d_m$), into specific predictions for binary classification or regression tasks.

To achieve this, we adopt a straightforward yet effective strategy of concatenating the output tensor, followed by the application of a linear layer. This approach, despite its simplicity, has demonstrated robust performance across our evaluated tasks, as detailed in the subsequent results section. The ability to utilize the entire sequence for prediction, rather than relying solely on the final token's representation (Luo et al. 2023), ensures a comprehensive integration of temporal features.

While this approach has demonstrated promising results, it is important to acknowledge the computational considerations this method entails. As the sequence length increases, the computational complexity of mapping a tensor from shape $(1, \text{sequence length} \times d_m)$ to the target shape escalates.

### Attention-based Prediction Head

Another approach leverages the attention mechanism to better capture the relationship between patch representations, potentially making the model more robust when dealing with long sequences. We compute attention scores among patch representations use them as coefficients to sum over the model output along the temporal dimension, so that the original model output tensor of dimension (sequence length, $d_m$) is reduced to a vector in $\mathbb{R}^{d_m}$. This final feature vector is used for downstream task prediction.

### Finetuning Process

In our finetuning framework, we synergize the objective loss $L_o$ with the signal modeling loss $L_m$ to formulate a comprehensive loss function: $L(y, y', X, X') = L_o(y, y') + \lambda L_m(X, X')$, where $X, y$ are PPG signals and labels of the dataset and $X', y'$ are model predictions of the signal and label. Empirically, we observed that this delivered much better performance and faster convergence compared to using objective loss only.

For regression tasks, we use the standard MSE as objective loss. Note that logit-Laplace loss does not apply here because the labels of the regression task (such as heart rate estimation) are not normalized to [0, 1]. For classification tasks, we use cross entropy as per standard. Signal modeling is done in the same way as it is in the pretraining phase, and hence uses logit-Laplace distribution loss.

A notable innovation in our finetuning process is the dynamic scheduling of the scaling parameter $\lambda$. While Radford et al. originally proposed to fix a scaling parameter $\lambda$ for the auxiliary loss, we found that it is more effective to use a dynamic scheduling technique that anneals $\lambda$ to 0.

## Method: Unsupervised Signal Denoising

Signal denoising is a critical preprocessing step in PPG signal analysis, aimed at enhancing signal quality and reliability for downstream tasks. We use the Segade model (Jain et al. 2023) to produce signal quality index sequence (SQI

| Task | Dataset | SOTA | GPT-19M | GPT-85M |
|---|---|---|---|---|
| **Blood Pressure Estimation** | Vital SBP (Wang et al. 2022) | 10.1 (Chowdhury et al. 2020) | 9.17 | **8.794** (Held-out) |
| | Vital DBP (Wang et al. 2022) | 7.64 (Chowdhury et al. 2020) | 12.5 | **7.119** (Held-out) |
| **Heart Rate Estimation** | WESAD (Schmidt et al., 2018) | **3.63** (Bieri et al. 2023) | 3.97 | 3.769/4.102 |
| | DaLiA (Reiss et al. 2019) | **2.61** (Bieri et al. 2023) | 1.92 | 3.102/1.887 |
| | IEEE (Zhang et al. 2015) | 3.18 (Bieri et al. 2023) | **2.97** | 2.986/3.220 |
| **AF Detection** | Stanford (Torres-Soto and Ashley. 2020) | 0.67 (Das et al. 2022) | 0.77 | **0.826/0.904** |
| **False Arrhythmia Alarm Detection** | False Alarm Real Time (Clifford et al. 2015) | **0.81** (Clifford et al. 2015) | 0.57 | 0.763/0.731 |
| | False Alarm Retrospective (Clifford et al. 2015) | **0.85** (Clifford et al. 2015) | 0.61 | 0.772/0.751 |
| **Respiration Rate** | BIDMC (Pimentel et al. 2016) | 1.51 (Kumar et al. 2022) | 1.91 | 1.791/**1.477** |

Table 1: Summary of Results
Heart rate and blood pressure estimation results are recorded in MAE, Stanford dataset recorded in F1, and false alarm datasets are recorded in custom metric.

sequence), which is the same length as the input signal. The SQI sequence is bounded within the interval [0, 1]. A data point is deemed "bad quality" if its SQI is below 0.5, and similarly, a patch is labeled as such if over 50% of its constituent points are of bad quality. In this denoising procedure, GPT is tasked with reconstructing patches identified as poor quality, while patches that are not flagged are kept as is to retain the original signal as much as possible.

While the generative application of GPT for denoising bypasses the need for explicit finetuning, it's worth noting the model's unidirectional nature as a limitation. This constraint may impede its performance, particularly when encountering low-quality patches early in the signal sequence, an issue more easily fixed when using bi-directional information like many current models do. Despite this, the exploratory nature of this work opens avenues for future investigations.

In moving forward, we aim to assess the impact of such denoising on downstream task performance and devise new ways to prompt the GPT model for higher reconstruction quality. To showcase the model's current prowess in signal recovery, we have included denoising examples in the appendix. For these demonstrations, we randomly select PPG samples outside the training set and mask out a certain percentage of the patches to mimic a noisy patch. We then ask the GPT to recover the original sequence. Through comparison with ground truth, we see that GPT can recover the original sequence faithfully even when up to half of it was compromised by artificial noise. However, the reconstruction quality notably diminishes when 70% of the signal is masked, a decline possibly linked to a higher concentration of masked patches early in the sequence.

## Results

Here we present finetuning results on various benchmarks. Our downstream tasks include heart rate estimation, atrial fibrillation detection, blood pressure estimation and detecting false arrhythmia alarms. The results reported come from the best between linear head prediction and attention head prediction.

Except for blood pressure estimation, all results reported in GPT-85M contains 2 entries. The first entry comes from a held-out test set, randomly sampled from the dataset, and the second entry comes from 5-fold cross validation. Because all downstream datasets are relatively small (except for BP estimation), cross validation result may be more valuable than that obtained from a randomly sampled test set.

In addition, we do not filter any bad quality signals when obtaining the results reported above. Due to different data preprocessing schemes, the table above only provides a reference.

## Conclusion and Future Work

This work presents a novel PPG foundation model. We keep the main component of GPT intact, while adapting the embedding and prediction layer to better suit the characteristics of PPG. We demonstrate that our trained model not only excels in a diverse set of downstream tasks but can also be used for generative tasks without further finetuning. We believe that similar techniques may be employed for other physiological signals and even continuous time series in general.

As a part of our future work, we plan to further investigate GPT's ability in generative downstream tasks and potential avenues in which it may better support other supervised tasks.

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

Sample reconstructions for 50% mask (noise)

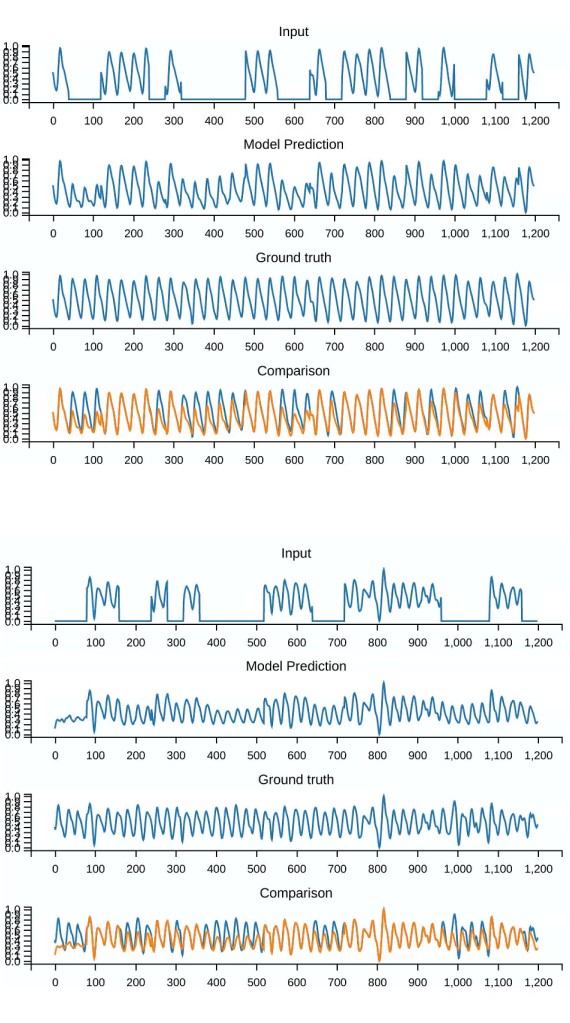

# Appendix

Sample reconstructions for 30% mask (noise)

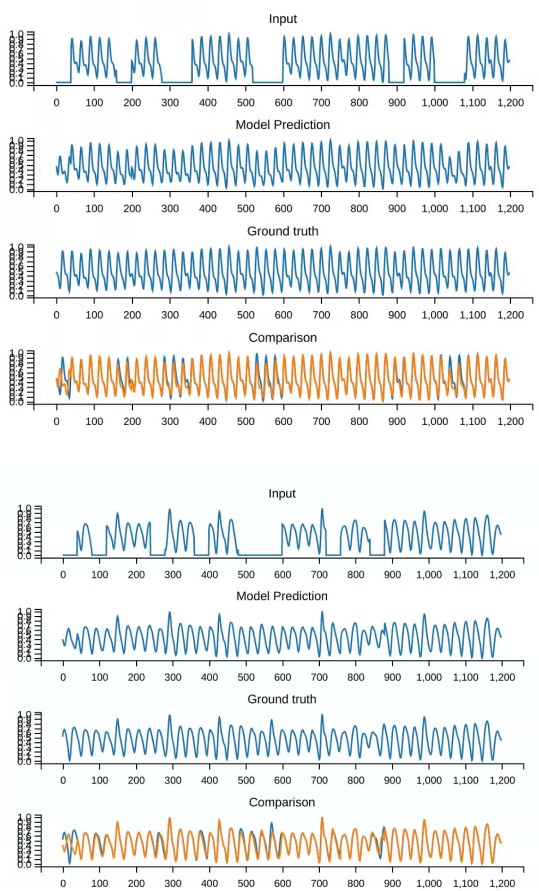

Sample reconstruction for 70% mask

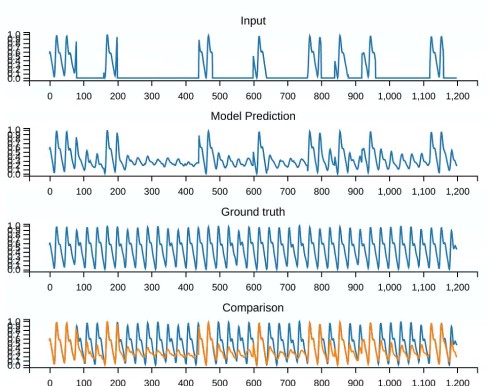