# OpenReview forum: "Adapting a Generative Pretrained Transformer Achieves SOTA Performance in Assessing Diverse Physiological Functions Using Only Photoplethysmography Signals: A GPT-PPG Approach"
_AAAI.org/2024/Spring_Symposium_Series/Clinical_FMs — AAAI 2024 SSS on Clinical FMs_

### Official Review · Reviewer_uhh3 · 2024-02-16

**Rating:** 5
**Confidence:** 4

**Review:**

Summary

This paper proposes architectural variants of transformers, pretraining, and fine-tuning methods for PPG domain.

Strengths

- Experiments are extensively conducted in PPG domain.

Weakness

- Lack of comparison with other transformer architectures for timeseries data
- Writing could be improved. For example, the difference between linear prediction head and attention-based prediction head is unclear, and it’s hard to identify SOTA algorithms in experiments.

---

### Official Review · Reviewer_gmkD · 2024-02-20
**Applying foundation model to an important domain.**

**Rating:** 7
**Confidence:** 2

**Review:**

Strength:
- This work develops PPG foundation models using a decoder-only transformer
- loss function, embedding, and linear head are specifically designed to fit PPG applications.
- Results in Table 1 show promising results

Weakness:
- Table 1 is a bit hard to read. Different performance metrics (MAE, F1, false alarm rates) are included. Please consider separating them.
- BP-SBP is not introduced. Also, why is 9.56 highlighted?
- In the conclusion section, it is claimed that the foundation model can be used for downstream tasks without further fine-tuning. I am not sure which experiments can support this claim.

---

### Official Review · Reviewer_MUNY · 2024-02-21
**GPT-PPG foundation model**

**Rating:** 7
**Confidence:** 4

**Review:**

Summary of Contributions:

The work proposes a Foundation model for (as the title suggests) “assessing diverse physiological functions, using only photoplethysmography signals”. The authors encode vast numbers of 30s of PPG samples @40Hz and train a GPT like foundation model with the next sample prediction pre-training objective. Then they fine-tune this model for solving downstream tasks such as heart rate estimation, atrial fibrillation detection, blood pressure estimation, and detecting false arrhythmia alarms.

Strengths:
1. The paper is well written and easy to follow. The block diagram is representative of the method presented in the paper.
2. The proposed method is sound and it stands to reason that with the increase in the model parameters and the training set, more emergent behaviour should be witnessed.
3. The qualitative results in the appendix are quite impressive.

Weaknesses:
1. Details such as number of model parameters, training time, GPUs used, etc. are missing from the paper. They often provide some indication on how scaling up might improve the results, and also about the feasibility of using the model.
2. The fine-tuning time requirements when compared to the training time (from scratch) of SOTA specialist models are also missing in the paper.
3. Some ablation studies are missing. For instance, it was mentioned that (in the decoder) the  RMSNorm was preferred over LayerNorm, then PoPE was used instead of Positional Encoding from the original transformers paper. Some indicator on how these choices would have affected the foundation model training might have been good. (Although the feasibility of these ablations will depend on the pre-training computational requirements, which again cannot be inferred unless disclosed in the paper)

---

### Official Review · Reviewer_fEni · 2024-02-21
**Interesting work**

**Rating:** 7
**Confidence:** 3

**Review:**

This paper applies the  Generative Pre-trained Transformer (GPT) model for photoplethysmography (PPG) signals. This is interesting and valuable. The point of using a logit-Laplace loss, instead of a MSE loss to train the model is also very insightful. My main concern about this paper is how can we make sure the converge when only using 5% of the data. We know transformers are data-hunger architectures. What is the limitation of such a model? As the paper mentioned in the future they will train the model using more data, I assume we will get the answer later. In general, this is an interesting and valuable paper for PPG-related tasks.